# Macrophage Cell Membrane Coating on Piperine-Loaded MIL-100(Fe) Nanoparticles for Breast Cancer Treatment

**DOI:** 10.3390/jfb14060319

**Published:** 2023-06-11

**Authors:** Christian Rafael Quijia, Geovana Navegante, Rafael Miguel Sábio, Valeria Valente, Alberto Ocaña, Carlos Alonso-Moreno, Regina Célia Galvão Frem, Marlus Chorilli

**Affiliations:** 1Department of Drugs and Medicines, School of Pharmaceutical Sciences, São Paulo State University (UNESP), Rodovia Araraquara Jau, Km 01-s/n-Campos Ville, Araraquara 14800-903, Brazil; christianqui47@gmail.com (C.R.Q.); rafael.m.sabio@unesp.br (R.M.S.); 2Department of Clinical Analysis, School of Pharmaceutical Sciences, São Paulo State University (UNESP), Rodovia Araraquara Jau, Km 01-s/n-Campos Ville, Araraquara 14800-903, Brazil; geonavegante@gmail.com (G.N.); valenteval@gmail.com (V.V.); 3Department of Medical Oncology, Hospital Clinico San Carlos and Health Research Institute of the Hospital Clinico San Carlos, 28040 Madrid, Spain; alberto.ocana@salud.madrid.org; 4Unidad NanoDrug, Facultad de Farmacia, Universidad de Castilla-La Mancha, 02008 Albacete, Spain; carlos.amoreno@uclm.es; 5Institute of Chemistry, São Paulo State University (UNESP), Prof. Francisco Degni 55, Araraquara 14800-060, Brazil; rcgfrem@gmail.com

**Keywords:** metal–organic framework, vesicle, cytotoxicity, nanostructures

## Abstract

Piperine (PIP), a compound found in *Piper longum*, has shown promise as a potential chemotherapeutic agent for breast cancer. However, its inherent toxicity has limited its application. To overcome this challenge, researchers have developed PIP@MIL-100(Fe), an organic metal–organic framework (MOF) that encapsulates PIP for breast cancer treatment. Nanotechnology offers further treatment options, including the modification of nanostructures with macrophage membranes (MM) to enhance the evasion of the immune system. In this study, the researchers aimed to evaluate the potential of MM-coated MOFs encapsulated with PIP for breast cancer treatment. They successfully synthesized MM@PIP@MIL-100(Fe) through impregnation synthesis. The presence of MM coating on the MOF surface was confirmed through SDS-PAGE analysis, which revealed distinct protein bands. Transmission electron microscopy (TEM) images demonstrated the existence of a PIP@MIL-100(Fe) core with a diameter of around 50 nm, surrounded by an outer lipid bilayer layer measuring approximately 10 nm in thickness. Furthermore, the researchers evaluated the cytotoxicity indices of the nanoparticles against various breast cancer cell lines, including MCF-7, BT-549, SKBR-3, and MDA. The results demonstrated that the MOFs exhibited between 4 and 17 times higher cytotoxicity (IC_50_) in all four cell lines compared to free PIP (IC_50_ = 193.67 ± 0.30 µM). These findings suggest that MM@PIP@MIL-100(Fe) holds potential as an effective treatment for breast cancer. The study’s outcomes highlight the potential of utilizing MM-coated MOFs encapsulated with PIP as an innovative approach for breast cancer therapy, offering improved cytotoxicity compared to free PIP alone. Further research and development are warranted to explore the clinical translation and optimize the efficacy and safety of this treatment strategy.

## 1. Introduction

Nanotechnology is a rapidly developing field that brings promising opportunities to human cancer diagnosis and treatment. Nanoparticles can play a significant role as a drug delivery system for cancer treatment [1]. This system can be classified into three groups: inorganic nanomaterials (including iron oxide, gold nanoparticles, and zeolites), organic nanomaterials (such as polymeric nanoparticles, micelles, dendrimers, and liposomes), and a third type known as metal–organic frameworks or porous coordination polymers (MOFs) [2]. MOFs are porous materials composed of metal ions or clusters of higher nuclearity and multifunctional organic ligands, enabling them to overcome the limitations of other nanostructured systems, such as low drug loading and release capacity [3]. MOFs can be synthesized using various methods that allow them to control their chemical and physical properties. The assembly of inorganic subunits and organic ligands through strong covalent ion bonds creates MOFs with high and regular porosities and adaptable porosity structures that can host molecules of different shapes [4].

Piperine (PIP) is an alkaloid extracted from seeds of plants in the Piperaceae family that has been used as an antimicrobial, antiparasitic, and antidepressant agent and as a modulator of oxidative stress-induced carcinogen [5]. Piperine has been demonstrated to inhibit the growth and survival of numerous different cancer cell types as well as to cause cell cycle arrest and apoptosis. It is especially potent against breast cancer and can target a variety of signaling pathways, oxidative stress, autophagy, and the activation of detoxifying enzymes [6,7]. However, its use presents difficulties related to its hydrophobic nature and high concentration requirement [8,9]. The incorporation of PIP into a nanostructured release system may help to overcome these difficulties. Various PIP-loaded nanostructures have been reported, such as carbon nanotubes, liposomes, and polymeric nanoparticles [10,11,12].

In a recent study conducted by Quijia et al. (2022) [13], PIP was encapsulated in MIL-100 (Fe). This MOF is a polycrystalline powder composed of iron (III) ions and 1,3,5-benzenetricarboxylic acid [14]. The main characteristics of this MOF (PIP@MIL-100 (Fe)) include particles measuring up to 120 nm in size and having a rhombohedral shape. It has a low polydispersity index of 0.03 and a hydrodynamic diameter of 98 ± 27.83 nm. Its zeta potential is +7 ± 0.6 mV. The nanosystem has a high loading capacity for PIP of 11.02% by weight (0.12 g g^−1^) and a high encapsulation efficiency of 95 ± 3%. MCF-7 and 4T1 breast cancer cell lines were used in cytotoxicity tests, which revealed that PIP@MIL-100 (Fe) had roughly three times the cytotoxicity (IC_50_) of free PIP. 

The surface modification strategies of MOF nanoparticles and materials engineering, such as the use of macrophage membranes, have the potential to improve the efficiency and selectivity of delivering chemotherapeutic agents and other biomedical applications. The functionalization of MOF nanoparticles improves their chemical and colloidal stability and offers the possibility of intravenous administration and interaction with specific receptors [15]. Moreover, the camouflaging of nanostructures with macrophage membranes can increase their blood circulation and reduce their clearance by mononuclear phagocyte systems, which can be useful in combating cancer and other diseases. These surface modification and materials engineering strategies are promising research areas in the search for more effective and selective therapies for various diseases [16,17,18].

Here, we present a comprehensive study involving the preparation and characterization of membrane vesicles and proteins extracted from macrophage cells (RAW 264.7 strain). Our objective was to develop a novel strategy for coating MIL-100 (Fe) with these biomolecules to achieve controlled release of PIP, a potent anti-cancer agent, against breast cancer. Previous studies have employed PIP for encapsulating these MOFs, as documented in published research [13]. To assess the efficacy and properties of our nanostructures, we employed a range of analytical techniques. The characterization included measurements of hydrodynamic diameter, polydispersity index (PdI), zeta potential, stability (evaluated on the first and tenth day after synthesis using dynamic light scattering), size (dry diameter), shape (observed through transmission electron microscopy (TEM) and scanning electron microscopy (SEM)), chemical behavior (infrared vibration spectroscopy (IR)), SDS-PAGE analysis, thermogravimetric analysis (TGA), and in vitro release kinetics of PIP.

Furthermore, to evaluate the effectiveness of our nanostructured systems, in vitro efficacy trials were conducted using breast cancer cells. These trials were intended to investigate the potential benefits of employing these innovative nanosystems for targeted drug delivery and therapy against breast cancer.

## 2. Materials and Methods

### 2.1. Materials

Piperine, trimesic acid (BTC), iron (III) chloride hexahydrate, 99% ethanol, methanol HPLC grade, and MTT (3-(4,5-dimethylthiazol-2-yl)-2,5-diphenyltetrazolium bromide) were purchased from Sigma-Aldrich (St. Louis, MO, USA). The cell culture medium used was RPMI-1640 supplemented with 10% fetal bovine serum (FBS) and 1% antibiotic (10,000 UI penicillin and 10 mg/mL of streptomycin solution) from Sigma Aldrich.

### 2.2. Methods

#### 2.2.1. Encapsulation of Piperine in MIL-100(Fe)

The PIP encapsulation process in MIL-100(Fe) was improved in comparison with the method described by Quijia et al. in 2022 [13], resulting in a drug encapsulation efficiency of 95 ± 3%. Additionally, the nanostructure was stored in absolute ethanol at 4 °C until use. In this study, we used the encapsulation percentage previously reported by our group, with an encapsulation efficiency of piperine in the MOFs of 95%, representing 0.025 mg/mg, which means mg of piperine per mg of MIL-100(Fe). 

A detailed description of the encapsulation method can be found in Appendix A. This section outlines step-by-step instructions, including the preparation of the encapsulation solution, incubation conditions, purification techniques, and any additional modifications or considerations involved in the process.

#### 2.2.2. Preparation of Macrophage Membrane Vesicles (MM)

To obtain MM-vesicles from RAW 264.7 cells, the method described by Gao et al. (2016) and Rao et al. (2017) [16,19] was employed. Macrophage cells were grown in culture flasks until they produced approximately 1 × 10^8^ cells mL^−1^. The cells were then extracted with 0.05% Trypsin-EDTA and centrifuged for 5 min at 1015 rpm at 4 °C. The resulting cells were homogenized using a portable Dounce homogenizer (20 passes on ice) and suspended in 10 mL of hypotonic lysis buffer composed of 20 mM of Tris-HCl, 10 mM of KCl, and 2 mM of MgCl (and a mini-protease inhibitor without EDTA) at 4 °C. The cell suspension was centrifuged at 5427 rpm for 5 min at 4 °C, and then, the supernatant was collected and centrifuged again for 30 min at 11,750 rpm (Figure 1A). The isolated membranes were dispersed in PBS (pH 7.4) at 4 °C for subsequent assays.

#### 2.2.3. Preparation of MM@PIP@MIL-100(Fe)

The MM@PIP@MIL-100(Fe) was prepared using the impregnation method, based on previous research [20,21,22,23,24], with some modifications. PIP@MIL-100 (Fe) (5.0 mg) was dispersed in 10 mL of PBS (pH = 7.4) and stirred for 30 min at 4 °C with previously extracted MM-vesicles (approximately 4 mg/mL). The functionalized MOFs were centrifuged at 10,000 rpm for 10 min, the supernatant was removed, and the nanoparticles were kept at 4 °C in PBS (pH = 7.4) (Figure 1B) [22,25].

#### 2.2.4. Photon Correlation Spectroscopy and Zeta Potential

The hydrodynamic diameter and zeta potential of nanoparticles dispersed in water at a concentration of 0.1 mg mL^−1^ were measured using an ultrasonic tip and the Zetasizer Nano series (Malvern Instruments Ltd., Malvern, UK), a product from the UK. Three readings were taken during the measurements, which were carried out at room temperature and under light scattering detection at an angle of 173°. The z-average size was determined using cumulant analysis with a repeatability of 1.6% using the Zetasizer Nano-ZS. The Nano software was used to convert the intensity distribution into volume using theoretical plots of the log of the relative scattering intensity versus particle size at angles of 173° [26,27].

#### 2.2.5. Infrared Vibrational Spectroscopy Analysis

A Perkin-Elmer 400 IR spectrometer (Perkin-Elmer Inc., Boston, MA, USA) was used to perform the vibrational spectroscopy analysis in the infrared region (IR) of the electromagnetic spectrum. An agate mortar was used to combine the samples with potassium bromide (KBr), and the mixture was then added to pellets for reading with a resolution of 2 cm^−1^. The method uses the molecules’ characteristic infrared radiation absorption to figure out the structures of the molecules.

#### 2.2.6. High-Resolution Transmission Electron Microscopy (HR-TEM) and High-Resolution Scanning Electron Microscopy (HR-SEM) 

The samples were coated with a thin layer of gold and mounted on a holder before HR-SEM was carried out on them using the TOPCON SM-300 microscope (Topcon Corporation, Hasunumachō, Japan) at 10–20 kV. Using a PHILIPS CM 200 SUPER TWIN transmission electron microscope (TEM), the morphology of purified nanosolids was examined, and photomicrographs were taken at various magnifications. The JEOL JEM-2100 (LaB6) (JEOL, Tokyo, Japan) at 100 kV, available from LME IQSC-USP in Sao Carlos, Brazil, was used for the TEM analyses. The samples were made by depositing 3 µL of diluted nanosuspensions on a copper grid, draining the excess liquid, drying it, and staining it for 3 min with 3 µL of 2% *w*/*v* aqueous uranyl acetate. The samples were dried after the excess stain was removed, and the staining process was then repeated. The grids were analyzed after drying at room temperature.

#### 2.2.7. Thermogravimetric Analysis (TGA)

TA Instruments’ TGA-Q500 instrument (TA Instruments, Delaware, United States) was used to perform the TGA analysis. A platinum pan with a maximum volume of 50 L was loaded with 5 mg of each sample and heated at a rate of 10 °C/min from 30 to 600 °C. For the analysis, a dry nitrogen flow of 40 mL/min was used to aid in the decomposition of MM@PIP@MIL-100(Fe) and PIP@MIL-100(Fe) [28].

#### 2.2.8. SDS-PAGE Characterization of MM@PIP@MIL-100(Fe)

The proteins were examined using sodium dodecyl sulfate–polyacrylamide gel electrophoresis (SDS-PAGE) [29]. Purified macrophage membrane vesicles (MM) and MM@PIP@MIL-100(Fe) were created in SDS sample buffer and measured with the BCA kit. Then, 20 g of the sample was loaded onto each well of a 10% SDS-PAGE after the samples were heated at 95 °C for 5 min. The samples were then run at 120 V for 2 h, and the resulting PAGE was stained for 2 h with Coomassie Blue and then washed overnight in preparation for visualization the next day on a gel documentation system [16].

#### 2.2.9. In Vitro Release Kinetics of PIP

Phosphate-buffered saline (PBS, pH = 7.4 or pH = 5) with 5% *v*/*v* Tween 20 and 5% *v*/*v* ethanol as the receptor medium was used to study the release kinetics of PIP under 37 °C two-dimensional agitation (agitation frequency = 150 rpm) [30]. The MM@PIP@MIL-100(Fe) nanosystem was first dissolved in 5 mL of the receptor medium at a concentration of 10 mg/mL. The samples were centrifuged, and a 1 mL sample of the supernatant was taken for PIP content analysis at predetermined time intervals (1–180 h). The medium was then replenished with 1 mL of the new receptor medium. To determine the amount of PIP released, the collected samples were analyzed using high-performance liquid chromatography (Appendix A). The add-in (DDSolver) for Microsoft Excel (Microsoft Excel, 2019) that offers statistical criteria to assess the model’s fitting quality was used to analyze the drug release model. To determine the best release model, two parameters—adjusted R^2^ and Akaike information criterion (AIC)—were evaluated while accounting for the volume of data, the number of data points, and the statistical analysis [31].

#### 2.2.10. Cell Viability Assay

MCF-7, MDA, SKBR-3, BT-549, and HaCaT cancer cells were put in 96-well plates at a density of 4000 cells/well and left there overnight. Dimethyl sulfoxide (DMSO) concentration in the study samples was kept under 0.25% (*v*/*v*) to prevent any negative effects on cell viability [32]. PIP was dissolved in DMSO at a concentration of 5% (*w*/*v*). DMEM medium was used to either disperse or dissolve MM@PIP@MIL-100 (Fe), which was then incubated for 48 h. In order to rule out any potential interference with the assay, nanoparticles in culture medium and culture medium alone were also tested along with the treatments, which were prepared at a 3 times higher concentration and added to the cells in a final volume of 200 µL per well.

#### 2.2.11. Statistical Analysis

The experiments were carried out three times (*n* = 3), and the mean SEM of the outcomes was obtained. Data comparison techniques included ANOVA with post-test or Student t-tests, with the significance set at *p* < 0.05. GraphPad Prism^®^ version 7.0 was used to conduct these statistical analyses. ANOVA was used to divide the observed variance data into various components for use in further tests. By obtaining the diameter distribution of the nanostructures and choosing the best fit for polydispersion based on a log-normal distribution, the modal diameter of the nanostructures was ascertained. The linear regression model based on the dose–response curve was used to calculate the inhibitory concentration (IC_50_) and its 95% confidence intervals.

## 3. Results and Discussion 

### 3.1. Analysis of Average Hydrodynamic Diameter and Zeta Potential

After coating the PIP@MIL-100(Fe) material with macrophage-derived vesicles (MM-vesicles) using the impregnation method (MM@PIP@MIL-100(Fe) material), DLS measurements were also performed. As expected, the results showed an increase in the average hydrodynamic diameter of the nanoparticles from 98 ± 27.83 to 150 ± 24.16 nm (Table 1), as well as a change in the zeta potential of the nanoparticles from +7 ± 0.6 to −32 ± 2.36 mV, suggesting that the membrane coating was successful. It is important to note that, under these circumstances, the acidic nature of the PIP@MIL-100(Fe) suspension (pH ~ 2.9–3.8) confers a positive charge to the material due to the protonation reaction of carboxylate groups. While the coating with the macrophage membrane vesicle (MM) has a pH of ~7.4, the surface of the MOFs acquires a negative charge due to electrostatic interactions between the positive ζ potential of the PIP@MIL-100(Fe) and the negative ζ potential of the MM (Table 1), due to the charge of the phospholipids and proteins constituting the cell membrane. Therefore, the acidic carboxylic groups of the MOF ligand help in the conjugation of primary amines or macromolecular groups, such as peptides and proteins from MM, ensuring the correct topological orientation of cell membranes in the MM@PIP@MIL-100(Fe) platform [2,25,33,34,35,36,37,38]. Table 1 summarizes the obtained results using the dynamic light scattering (DLS) technique on the Zetasizer Nano ZS equipment, including the average hydrodynamic diameter, polydispersity index (PdI), and zeta potential of the nanoparticles.

### 3.2. Analysis of Nanostructured Systems by Vibrational Spectroscopy in the Infrared Region (FT-IR)

The coating with MM on the MOF exhibited absorption bands similar to those of the cell membrane vesicles previously reported, such as the 1800–1350 cm^−1^ lipid ester groups and amide I and II protein bands (Figure 2) [39]. The vibrational spectrum between 1100 and 800 cm^−1^ and the stretching and bending of phosphate groups provide some information about the stretching and bending bands of the phosphate groups. The strong band ν_1_ (PO_4_) typically belongs to the region of 1000–1100 cm^−1^, while the bending mode ν_2_ (PO_4_) usually appears as a medium to strong band in the range of 600–900 cm^−1^. Additionally, the symmetric stretching (v_1_) of the phosphate groups (PO_4_^3−^) is typically exhibited around 1050–1100 cm^−1^ [40]. This confirms the presence of vesicles in the MOFs, including both protein and lipid molecules.

### 3.3. Morphology Analysis of Nanomaterials

In this study, the morphology and particle size of materials based on MOF were prepared and analyzed using scanning electron microscopy (SEM) and transmission electron microscopy (TEM) techniques.

Uranyl acetate was used to negatively stain the MM@PIP@MIL-100(Fe) material’s particles, which were then visualized using TEM. The images produced using the technique clearly displayed a PIP@MIL-100(Fe) core with a diameter of about 50 nm. As shown in Figure 3D, there was an additional lipid bilayer outer layer that was 10 nm thick. This demonstrated that the cell membrane had successfully been coated onto the PIP@MIL-100(Fe) composite. Additionally, Figure 3A,C show the SEM images for PIP@MIL-100(Fe) and MM@PIP@MIL-100(Fe), respectively. 

### 3.4. Thermogravimetric Analysis 

The thermal behavior of the PIP@MIL-100(Fe) and MM@PIP@MIL-100(Fe) materials was studied using thermogravimetric analysis (TGA). The analysis was conducted with a heating rate of 10 °C/min, and the temperatures ranged from 30 °C to 600 °C. The thermogravimetric curves of the materials used in this study are illustrated in Figure 4.

In the thermal decomposition of the MM coating in MOFs (red line in Figure 4), the sample in the region between 25 and 73 °C showed a mass loss of 19.4% associated with the coating of macrophage membrane vesicles (MM) of MM@PIP@MIL-100(Fe). These values are likely due to the release of moisture, mainly from the cellular vesicle or the heating of the organic matter [41].

### 3.5. Characterization and Stability of the MM@PIP@MIL-100(Fe) Platform

SDS-PAGE was used to analyze the proteins from MM-vesicles and the purified MM@PIP@MIL-100(Fe) materials. Compared to natural MM-vesicles, the proteins were successfully coated onto MM@PIP@MIL-100(Fe) after the impregnation treatment, according to the results shown in Figure 5A. This suggests that proteins were successfully transferred from the macrophage’s natural membrane to the PIP@MIL-100(Fe) composite. MM@PIP@MIL-100(Fe) stability was also examined, and samples kept in 1 PBS (4 °C) for 10 days revealed no discernible size change (*p*-value > 0.05) (Figure 5B). This finding indicates that the nanoparticles may still maintain their stable structure, which is important for upcoming biomedical research.

### 3.6. In Vitro Release Assay of PIP

To examine the release profile of PIP, a study was carried out under physiologically relevant conditions in a phosphate-buffered saline (PBS) solution at both pH 7.4 and 5.0, with the addition of 5% *v*/*v* Tween 20 and 5% *v*/*v* ethanol, at a temperature of 37 °C. These conditions were selected to simulate both the blood circulation and the acidic microenvironments present in tumor regions [42]. Because most nanostructures tend to be internalized by endocytosis in cancer cells and become trapped in endosomal and lysosomal compartments, which typically range in pH from 4.5 to 5.5, pH 5.0 was chosen. [43].

Table 2 shows that the release of PIP from MM@PIP@MIL-100(Fe) followed the simplified Korsmeyer–Peppas model with a high R^2^ (>0.95) and n values > 0.43, indicating that the release mechanism was governed by diffusion. The release curve exhibited a slow stage with only about 25% (0.15 mg mL^−1^) of PIP released at pH = 5 after 14 days. For pH = 7.4, the release was even slower, with only 7.2% (0.014 mg mL^−1^) released (Figure 6). 

According to Palanikumar et al. (2022) [44], the presence of MM-vesicles impedes drug release from the PIP@MIL-100(Fe) material, although it does not entirely eliminate it. As a result, the MM@PIP@MIL-100(Fe) platform has a highly stable encapsulation, which is essential in preventing premature release and guaranteeing that loaded drugs eventually reach target cancer cells. Additionally, the decrease in release at pH = 7.4 is likely attributed to the safeguarding function of MM-vesicles on the nanomaterial. This finding indicates the potential for PIP to have a controlled release from pH-sensitive MM-MIL-100(Fe), which is loaded with PIP in acidic solutions. This feature is important as it minimizes side effects while increasing PIP accumulation at the tumor sites [45].

On the other hand, the release of PIP at pH 7.4 (7%) is much lower compared to the amount released at pH 5.0 (25%) after 120 h of analysis (Figure 6), which is likely due to cell membrane vesicles. The vesicles have relatively alkaline pH values (pH 7.4–7.1); in acidic pH, there are changes in the electrical charge of the membrane from groups present in the lipid molecule and peptides, causing the degradation or rupture of the membrane [46,47,48,49].

### 3.7. Cytotoxicity

Table 3 and Appendix A, in parentheses, present the number of times that the nanostructures exceeded the IC_50_ in relation to the drug (PIP). MM@PIP@MIL-100 (Fe) exhibited high cytotoxicity (IC_50_) in the four cell lines compared to the free piperine. All PIP-loaded nanosystems had a more significant toxic effect (*p* < 0.05) on tumor cells than on free PIP, according to the statistical analysis using Tukey’s ANOVA test in comparison with the IC_50_ (Figure 7). Lysosomal enzymes with an acidic pH are probably the cause of this cytotoxic effect because they can significantly increase drug release in the cytoplasm and cause cell death [50].

It is important to acknowledge that the coating of MM-vesicles onto MOFs exhibited higher toxicity in certain cell lines. A study by Wuttke et al. (2015) demonstrated that the coating of nanoMOFs with lipid bilayers can enhance their uptake by cancer cells [22]. Additionally, previous research has indicated that macrophage membranes possess the ability to actively bind to cancer cells due to the high expression of α4 and β1 integrins in RAW 264.7 cells, providing them with specific metastasis-targeting capabilities [51]. Consequently, the presence of these proteins in MOFs contributed to improved absorption and increased toxicity in cancer cells.

## 4. Conclusions

In this study, the researchers successfully coated PIP@MIL-100(Fe) with MM-vesicles, resulting in the creation of a new material called MM@PIP@MIL-100(Fe). The coating process was confirmed using various analyses and measurements. DLS measurements indicated an increase in the average hydrodynamic diameter of the nanoparticles, along with a change in their zeta potential, providing evidence of successful coating. TEM imaging revealed the presence of a PIP@MIL-100(Fe) core with a diameter of approximately 50 nm, surrounded by an outer layer of lipid bilayer measuring about 10 nm in thickness, indicating successful cell membrane coating. TGA was conducted to investigate the thermal behavior of the materials, while SDS-PAGE analysis was utilized to analyze the proteins coated onto MM@PIP@MIL-100(Fe) following the impregnation treatment. Stability assays demonstrated that the nanoparticles maintained a stable structure. The release of PIP from MM@PIP@MIL-100(Fe) was characterized and found to exhibit a high correlation with the simplified Korsmeyer–Peppas model, suggesting controlled and predictable release kinetics. Overall, these findings indicate that MM@PIP@MIL-100(Fe) has potential applications in future biomedical analyses, highlighting its suitability for further research in the field. Moreover, additional research is needed to explore these potential applications in more detail and to conduct in vivo studies to validate the efficacy and safety of MM@PIP@MIL-100(Fe) for various biomedical applications. These future directions will contribute to the advancement of nanomedicine and the development of innovative therapeutic and diagnostic strategies.

## Figures and Tables

**Figure 1 jfb-14-00319-f001:**
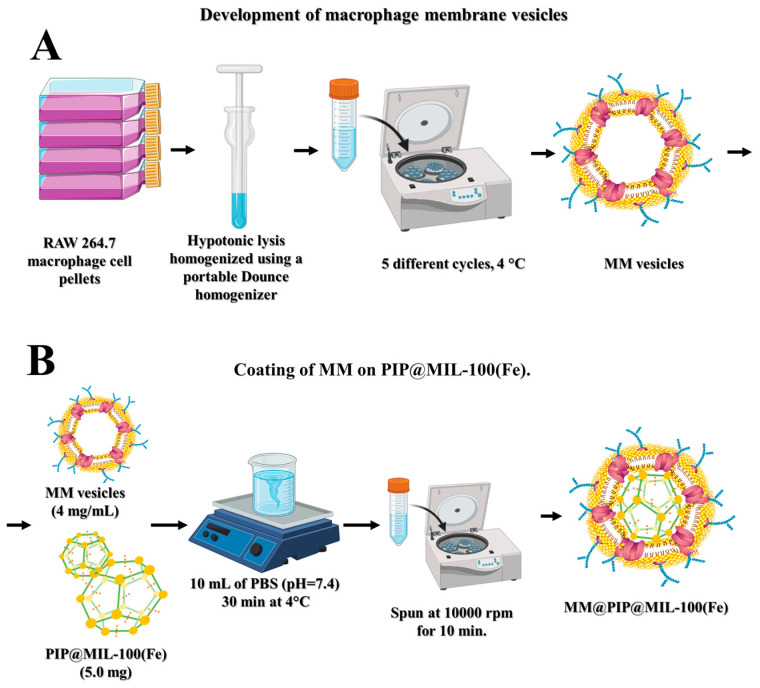
(**A**). Development of macrophage membrane vesicles. (**B**). Development of MM@PIP@MIL-100(Fe) nanostructures.

**Figure 2 jfb-14-00319-f002:**
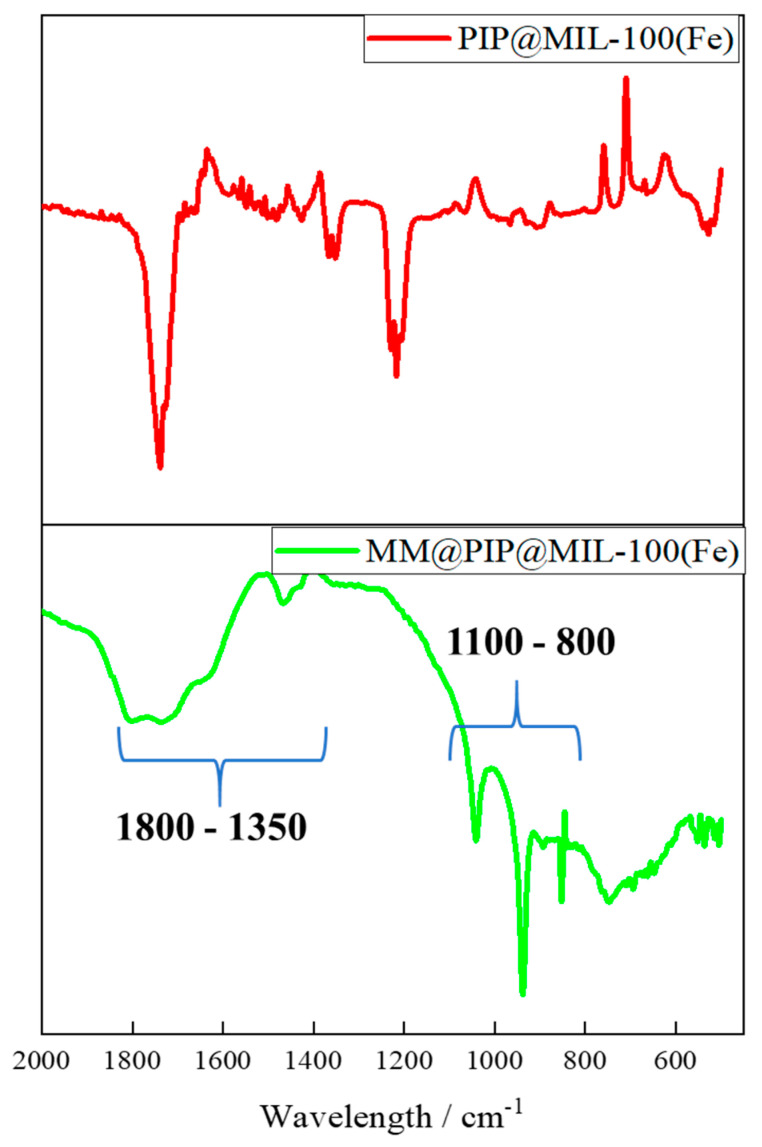
Fourier-transform infrared (FT-IR) spectrum analysis of nanoparticles: PIP@MIL-100(Fe) (red line); MM@PIP@MIL-100(Fe) (green line).

**Figure 3 jfb-14-00319-f003:**
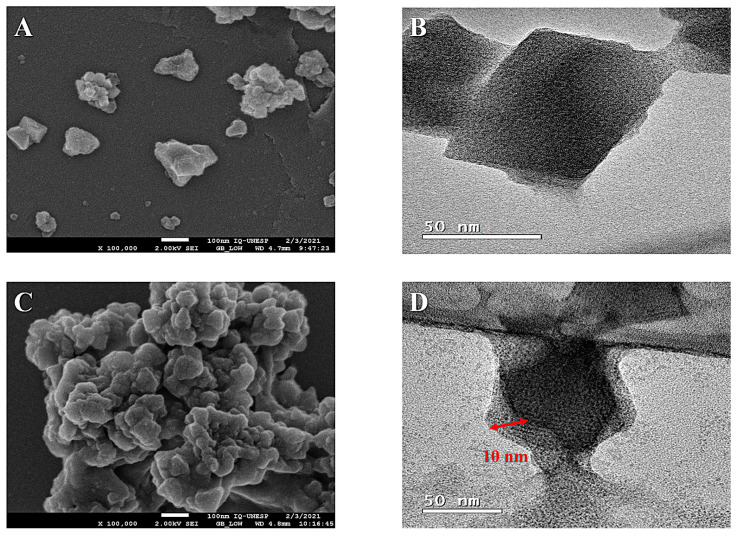
Morphology analysis using scanning electron microscopy (SEM): (**A**) PIP@MIL-100 (Fe); (**C**) MM@PIP@MIL-100 (Fe). Morphology analysis using transmission electron microscopy (TEM): (**B**) PIP@MIL-100(Fe) and (**D**) MM@PIP@MIL-100(Fe). Note: (**A**,**B**) are from a previous article by Quijia et al. (2022) [13].

**Figure 4 jfb-14-00319-f004:**
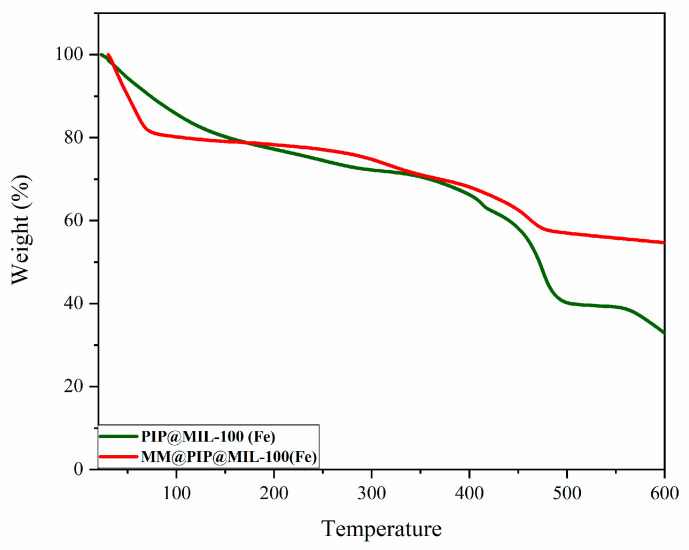
Thermogravimetric curves of the compounds: PIP@MIL-100(Fe) (green line); MM@PIP@MIL-100(Fe) (red line).

**Figure 5 jfb-14-00319-f005:**
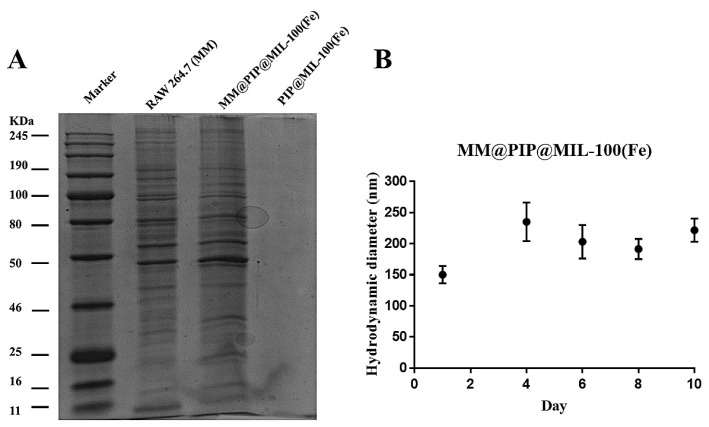
(**A**) SDS-PAGE analyses of proteins extracted from the membrane of RAW 264.7 cell line and MM@PIP@MIL-100(Fe). (**B**) Verification of the average hydrodynamic diameter of MM@PIP@MIL-100(Fe) sample stored in 1× PBS (4 °C) for 10 days. Data with standard deviation (n = 3), ANOVA with Tukey’s post-test, *p* < 0.05.

**Figure 6 jfb-14-00319-f006:**
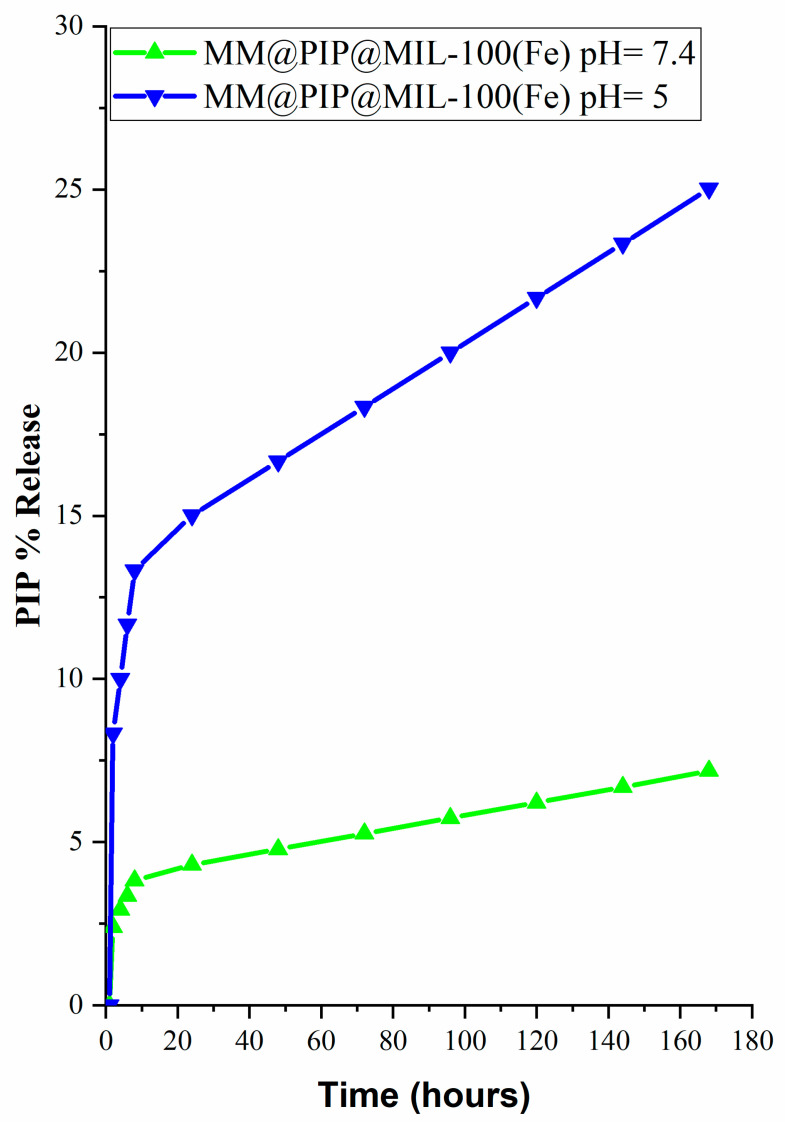
Release profile of MM@PIP@MIL-100(Fe) at pH values 7.4 and 5.0.

**Figure 7 jfb-14-00319-f007:**
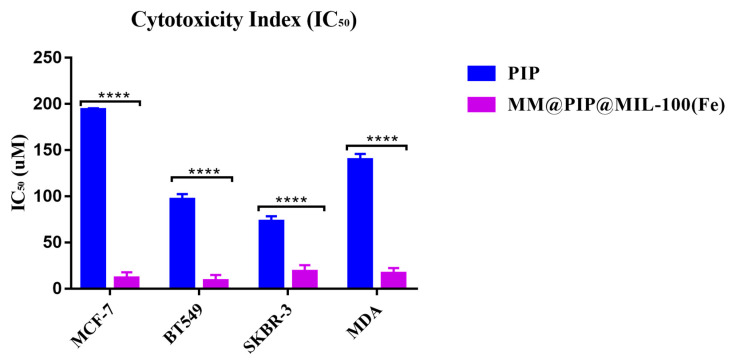
Cell viability of MCF-7; BT-549; SKBR-3 and MDA were treated with 6.25 to 100 μM and incubated for 48 h. Piperine (PIP), macrophage membranes (MM). The results represented in the figure refer to the averages of three independent experiments (mean ± standard deviation). IC_50_ corresponds to the minimum concentration to inhibit 50% of cancer cells. Asterisk indicates statistical significance **** *p* < 0.0001.

**Table 1 jfb-14-00319-t001:** Zeta potential (ζ) analysis, hydrodynamic diameter, and polydispersity index of different nanostructures.

	PIP@MIL-100 (Fe)	Vesicle (MM)	MM@PIP@MIL-100(Fe)
Zeta Potential (mV)	+7 ± 0.6	−14 ± 1.50	−32 ± 2.36
Hydrodynamic diameter (nm)	98 ± 27.83	88 ± 0.81	150 ± 24.16
Polydispersity index	0.03 ± 0.006	0.4 ± 0.09	0.4 ± 0.05

**Table 2 jfb-14-00319-t002:** Fitting of the release profile equation, where M is the cumulative release (%) and t is the release time (h).

	MM@PIP@MIL-100(Fe)	
	pH = 7.4	pH = 5.0
Korsmeyer–Peppas Model	$M=k_{KP}*t^{n}$	
Equation	M = 2.17 t^0.21^	M = 7.51 t^0.22^
Rsqr	0.97221752	0.972872357
AIC	0.000	0.027
Weibull Model	$M=100\left[e^{-\frac{{\left(t-Ti\right)}^{\beta }}{\alpha }}\right]$	
Equation	$M=100\left[e^{-\frac{{\left(t-0.8\right)}^{0.20}}{42.18}}\right]$	$M=100\left[e^{-\frac{{\left(t-0.8\right)}^{0.22}}{11.93}}\right]$
Rsqr	0.96896075	0.967943034
AIC	0.003	0.031
Gompertz Model	$M=100*e^{-\alpha *e^{-\beta *\log(t)}}$	
Equation	$M=100*e^{-0.004*e^{-0.000*\log(t)}}$	$M=100*e^{-0.003*e^{-0.000*\log(t)}}$
Rsqr	0.96557946	0.96128035
AIC	0.002	0.031

Note: Adjusted R^2^ (Rsqr) and the Akaike information criterion (AIC).

**Table 3 jfb-14-00319-t003:** Cytotoxicity index of nanoparticles and piperine against breast cancer cells.

Cytotoxicity Index (IC_50_) Expressed in μM				
	MCF-7	MDA	SKBR-3	BT-549
PIP	193.67 ± 0.30	139.60 ± 1.17	72.62 ± 1.08	96.38 ± 1.10
MM@PIP@MIL-100 (Fe)	11.45 ± 1.18 (17)	16.32 ± 1.12 (8)	18.51 ± 1.29 (4)	8.71 ± 1.12 (12)

Note: Piperine (PIP), macrophage membranes (MM). The results represented in the table refer to the averages of three independent experiments (mean ± standard deviation). IC_50_ corresponds to the minimum concentration to inhibit 50% of cancer cells. Cells were treated with 6.25 to 100 μM and incubated for 48 h. The number of times the nanostructures exceeded the IC_50_ in relation to the drug (PIP) is shown in parentheses.

## Data Availability

Not applicable.

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
