# Peer review of "Macrophage Cell Membrane Coating on Piperine-Loaded MIL-100(Fe) Nanoparticles for Breast Cancer Treatment"

_jfb, 2023, doi:10.3390/jfb14060319_

Round 1

Reviewer 1 Report

Overall the manuscript is a nice effort and merit publishing. However i will suggest these changes before further processing.

1. In the abstract descript methods and add some crispy results not just statements.

2. Better to start the introduction from the nanotechnology not piperine. It can be added in the second section

3. The references are not as per journal recommended style

4. Improve the resolution of images

5. Images of cells viability and flow cytometry are required

6. What is the unit of cytotoxicity in table 3

7. Carefully check typos and possible mistakes in the manuscript

Minor but careful langue editing is required before publishing. 

Author Response

We make changes to the text.

Point by point reply to the reviewers' comments of the manuscript: jfb-2422138 (“Macrophage Cell Membrane Coating on Piperine-Loaded MIL-100(Fe) Nanoparticles for Breast Cancer Treatment.”)

We would like to thank again the reviewers for their constructive criticism and helpful suggestions. Below, please find our point-by-point replies to their comments, written in blue after each reviewer’s comments/suggestions.

In second new version of the manuscript, all recent modifications are highlighted in light blue.

1.- In the abstract descript methods and add some crispy results not just statements.

  • We made modifications to the abstract of the article with more details (Lines 22-41).

2.- Better to start the introduction from the nanotechnology not piperine. It can be added in the second section.

  • This has been modified.

3.- The references are not as per journal recommended style.

  • This has been modified.

4.- Improve the resolution of images

  • The resolution of the images has been improved.

5.- Images of cells viability and flow cytometry are required

  • Cell viability has been included in the supplementary file. However, it was not possible to perform the flow cytometry study due to experimental conditions. These conditions were caused by the nanosystems themselves, as nanoparticle clusters formed during the cytometer analysis, hindering the analysis.

6.- What is the unit of cytotoxicity in table 3

  • The unit of cytotoxicity in Table 3 has been added (Line 375).
  1. Carefully check typos and possible mistakes in the manuscript
  • Typographical and English errors have been corrected.

Reviewer 2 Report

This manuscript developed PIP@MIL-100(Fe) as an appealing platform for the treatment of breast cancer. The authors fully characterize their nanoparticles. SDS-PAGE analysis confirmed the presence of proteins on the MOF surface. This study suggests that MM-coated MOFs encapsulated with PIP have the potential to be an effective treatment for breast cancer. All the characterizations are properly done and well support the conclusions.

Considering the novelty of this manuscript and the quality of the data, I believe this work may suitable for this journal. However, the current form frowns on the direct publication and there’re a lot points needed to be addressed:

(1)    For the in vitro release kinetics of PIP, the pH of PBS is 7.4 or 7.0? The data of Table 2 is inconsistent with Figure 6.

(2)    For the FT-IR of PIP@MIL-100(Fe), it should be performed again. Please confirm phosphate groups is PO43- or PO2-.

(3)    How about the release cycle of these nanoparticles?

The English is fine. There're some minor wording issues.

Author Response

We make changes to the text.

Point by point reply to the reviewers' comments of the manuscript: jfb-2422138 (“Macrophage Cell Membrane Coating on Piperine-Loaded MIL-100(Fe) Nanoparticles for Breast Cancer Treatment.”)

We would like to thank again the reviewers for their constructive criticism and helpful suggestions. Below, please find our point-by-point replies to their comments, written in blue after each reviewer’s comments/suggestions.

In second new version of the manuscript, all recent modifications are highlighted in light blue.

(1)    For the in vitro release kinetics of PIP, the pH of PBS is 7.4 or 7.0? The data of Table 2 is inconsistent with Figure 6.

  • It was a typographical error.

(2)    For the FT-IR of PIP@MIL-100(Fe), it should be performed again. Please confirm phosphate groups is PO43- or PO2-.

  • The FT-IR data of PIP@MIL-100(Fe) were analyzed multiple times for the publication, demonstrating consistent results.
  • Include information about the phosphate groups and compare the data presented in the article with the existing literature. (Line 253-259)

 (3)    How about the release cycle of these nanoparticles?

  • The MIL-100(Fe) nanoparticle has been previously mentioned in previous studies, which demonstrate that these particles undergo a gradual degradation in simulated biological environments [1]. Research has been conducted on the degradation kinetics of MIL-100(Fe) nanoparticles at 37 °C in water and PBS, using X-ray absorption near-edge structure spectroscopy (XANES) to analyze the degradation mechanism. [2]The results conclude that these nanoparticles exhibit a degradation rate in the presence of PBS or other biological conditions, implying the release of encapsulated drugs. For instance, previous studies have shown that approximately 93% of the drugs are released within a 48-hour period for MIL-100(Fe) nanoparticles. [3]In our study, we have confirmed drug release using the same material, as discussed in our previous article. [4]

Reference Bibliographic

[1]        H. J. C. Bunzen, "Chemical stability of metal‐organic frameworks for applications in drug delivery," vol. 7, no. 9, pp. 998-1007, 2021.

[2]        C. R. Quijia et al., "Application of MIL-100 (Fe) in drug delivery and biomedicine," vol. 61, p. 102217, 2021.

[3]        M. P. Abuçafy et al., "MIL-100 (Fe) Sub-Micrometric Capsules as a Dual Drug Delivery System," vol. 23, no. 14, p. 7670, 2022.

[4]        C. R. Quijia et al., "In situ synthesis of piperine-loaded MIL-100 (Fe) in microwave for breast cancer treatment," vol. 75, p. 103718, 2022.

Reviewer 3 Report

The article titled "Macrophage Cell Membrane Coating on Piperine-Loaded MIL-100(Fe) Nanoparticles for Breast Cancer Treatment" explores an interesting new application of MIL-100(Fe) for the treatment of breast cancer. The authors have conducted comprehensive analytical characterizations and in vitro studies, and their results are very promising. However, the authors should address the below points before their manuscript can be accepted for publication.

 1.      Authors should provide N2 sorption data for all of the MOFs (PIP@MIL-100(Fe), MM@PIP@MIL-100(Fe), and pristine MIL-100(Fe)). This data would help to characterize the MOFs further and to understand how the presence of piperine and macrophage cell membranes affects their properties.

2.      The authors should quantify how much piperine is loaded into the MOFs and how this loading affects the release of piperine from the MOFs. This information would be important for understanding the potential therapeutic efficacy of MOFs.

3.      They should determine the kinetic diameter of piperine. This information would be important for understanding how piperine interacts with the MOFs and cancer cells.

4.      Authors should investigate whether Fe leaching occurs from the MOFs during the release of piperine. If Fe leaching does occur, the authors should determine the extent of the leaching and how it affects the therapeutic efficacy of the MOFs.

5.      Should assign the stretching frequencies of MM and piperine in the IR spectra.

6.      The Manuscript should discuss the stability of the MOFs in the presence of biological fluids in a specific paragraph. This information is important for understanding the potential toxicity of MOFs and for developing strategies to improve their stability. 

Author Response

We make changes to the text.

Point by point reply to the reviewers' comments of the manuscript: jfb-2422138 (“Macrophage Cell Membrane Coating on Piperine-Loaded MIL-100(Fe) Nanoparticles for Breast Cancer Treatment.”)

We would like to thank again the reviewers for their constructive criticism and helpful suggestions. Below, please find our point-by-point replies to their comments, written in blue after each reviewer’s comments/suggestions.

In second new version of the manuscript, all recent modifications are highlighted in light blue.

  1. Authors should provide N2 sorption data for all of the MOFs (PIP@MIL-100(Fe), MM@PIP@MIL-100(Fe), and pristine MIL-100(Fe)). This data would help to characterize the MOFs further and to understand how the presence of piperine and macrophage cell membranes affects their properties.
  • Thank you very much for your suggestion. During the development of the research, we also had the objective of analyzing the absorption of N2. Unfortunately, we were unable to perform this study due to the amount required for analysis, which was approximately 40 mg of MM@PIP@MIL-100(Fe). Also, we needed large numbers of cell vesicles, so we did not do those studies. However, our group has already carried out an exhaustive study in relation to the PIP encapsulated in MIL-100 (Fe), where we did analyze the adsorption. [1]
  1. The authors should quantify how much piperine is loaded into the MOFs and how this loading affects the release of piperine from the MOFs. This information would be important for understanding the potential therapeutic efficacy of MOFs.
  • In this study, the encapsulation percentage previously reported in our group was used, where the encapsulation efficiency of piperine in the MOFs was 95%, representing 0.025 mg/mg, which means mg of piperine per mg of MIL-100(Fe). (Line 118-124)
  • Therefore, regarding whether the loading affects the release, it does not influence it because the drug was first encapsulated in situ and then coated with the membrane, which does not affect the amount of drug inside.
  1. They should determine the kinetic diameter of piperine. This information would be important for understanding how piperine interacts with the MOFs and cancer cells.
  • While determining the kinetic diameter of piperine would provide valuable information about its interaction with MOFs and cancer cells, we have not conducted this study due to challenges or limitations. However, we will consider and carry out this research for future investigations.
  1. Authors should investigate whether Fe leaching occurs from the MOFs during the release of piperine. If Fe leaching does occur, the authors should determine the extent of the leaching and how it affects the therapeutic efficacy of the MOFs.
  • Many studies have been conducted on MIL-100(Fe) in the field of medicine. [2]These studies have shown that this MOF does not cause cytotoxicity in quantities less than 1.2 mg/mL, which was significantly lower than the amount used in our study. Although we have investigated the existence of studies on the leaching of Fe from MIL-100(Fe), we have found no publications on this topic. However, there are studies available regarding the stability of the MOF under reaction conditions. [3-7]
  1. Should assign the stretching frequencies of MM and piperine in the IR spectra.
  • Regarding the analysis of infrared (IR) spectroscopy of the cell membrane, we were unable to carry it out due to the required quantity of cellular vesicles. Therefore, we conducted other studies in which it was not necessary to use a large amount of cell membrane. As for piperine, we have attached the results of the IR spectroscopy in a supporting file included with the article.

  1. The Manuscript should discuss the stability of the MOFs in the presence of biological fluids in a specific paragraph. This information is important for understanding the potential toxicity of MOFs and for developing strategies to improve their stability.
  • The stability of MIL-100(Fe) in biological fluid media has been extensively studied, which is why we did not conduct this study.[8, 9] However, regarding the interaction between MOFs and MM, it would be interesting to explore this issue, although it was not the main objective of the scientific article. As previously mentioned, we plan to conduct further studies on this MOFs in future research and will consider adding this suggestion to our investigation.

Reference Bibliographic

[1]        C. R. Quijia et al., "In situ synthesis of piperine-loaded MIL-100 (Fe) in microwave for breast cancer treatment," vol. 75, p. 103718, 2022.

[2]        C. R. Quijia et al., "Application of MIL-100 (Fe) in drug delivery and biomedicine," vol. 61, p. 102217, 2021.

[3]        R. Nivetha et al., "Highly porous MIL-100 (Fe) for the hydrogen evolution reaction (HER) in acidic and basic media," vol. 5, no. 30, pp. 18941-18949, 2020.

[4]        A. Dhakshinamoorthy et al., "Comparison of porous iron trimesates basolite F300 and MIL-100 (Fe) as heterogeneous catalysts for lewis acid and oxidation reactions: roles of structural defects and stability," vol. 2, no. 10, pp. 2060-2065, 2012.

[5]        J. J. Delgado-Marín, J. Narciso, and E. V. J. M. Ramos-Fernández, "Effect of the Synthesis Conditions of MIL-100 (Fe) on Its Catalytic Properties and Stability under Reaction Conditions," vol. 15, no. 18, p. 6499, 2022.

[6]        H. Lv, H. Zhao, T. Cao, L. Qian, Y. Wang, and G. J. J. o. M. C. A. C. Zhao, "Efficient degradation of high concentration azo-dye wastewater by heterogeneous Fenton process with iron-based metal-organic framework," vol. 400, pp. 81-89, 2015.

[7]        M. Giménez-Marqués et al., "Exploring the catalytic performance of a series of bimetallic MIL-100 (Fe, Ni) MOFs," vol. 7, no. 35, pp. 20285-20292, 2019.

[8]        X. Li et al., "New insights into the degradation mechanism of metal-organic frameworks drug carriers," Scientific reports, vol. 7, no. 1, p. 13142, 2017.

[9]        E. Bellido, M. Guillevic, T. Hidalgo, M. J. Santander-Ortega, C. Serre, and P. J. L. Horcajada, "Understanding the colloidal stability of the mesoporous MIL-100 (Fe) nanoparticles in physiological media," Langmuir, vol. 30, no. 20, pp. 5911-5920, 2014.